# Paracetamol Intake and Hematologic Malignancies: A Meta-Analysis of Observational Studies

**DOI:** 10.3390/jcm10112429

**Published:** 2021-05-30

**Authors:** Jesús Prego-Domínguez, Bahi Takkouche

**Affiliations:** 1Department of Preventive Medicine, University of Santiago de Compostela, 15782 Santiago de Compostela, Spain; susofis@gmail.com; 2Centro de Investigación Biomédica en Red de Epidemiología y Salud Pública (CIBER-ESP), 28029 Madrid, Spain

**Keywords:** paracetamol, hematologic neoplasms, meta-analysis

## Abstract

Hematologic malignancies cause more than half a million deaths every year worldwide. Analgesics were suggested as chemopreventive agents for several cancers but so far, results from individual studies about the relationship between paracetamol (acetaminophen) use and hematologic malignancies are conflicting. Therefore, we decided to perform a systematic review and meta-analysis. We retrieved studies published in any language by systematically searching Medline, Embase, Conference Proceedings Citation Index, Open Access Theses and Dissertations, and the five regional bibliographic databases of the World Health Organization until December 2020. Pooled odds ratios (OR) and their 95% confidence intervals (CI) were calculated according to the inverse of their variances. We performed separate analyses by histologic type. We also evaluated publication bias and assessed quality. A total of 17 study units met our inclusion criteria. The results show an association of hematologic malignancies with any paracetamol intake (OR 1.49, 95% CI 1.23–1.80) and with high paracetamol intake (OR 1.77, 95% CI 1.45–2.16). By subtype, risk was higher for multiple myeloma (OR 2.13, 95% CI 1.54–2.94) for any use and OR 3.16, 95% CI 1.96–5.10 for high intake, while risk was lower and non-significant for non-Hodgkin lymphoma. This meta-analysis provides evidence that paracetamol intake may be associated with hematologic malignancies and suggests that a dose–response effect is plausible. These results are unlikely to be due to publication bias or low quality of studies. Future research should focus on assessing the dose–response relationship.

## 1. Introduction

Hematologic malignancies are a heterogeneous group of diseases of diverse incidence, prognosis, and etiology that cause 1.3 million cases and more than 700,000 deaths every year worldwide [1]. These malignancies show a stable trend in the last decades, but an increased incidence is observed for myeloma in women [2].

Analgesics, mostly aspirin and non-steroidal anti-inflammatory drugs (NSAIDs) were proposed as chemopreventive agents in cancers of several anatomic locations, including colon and breast, due to their cyclooxygenase (COX) prostaglandin inhibition [3,4]. Prostaglandins are known to inhibit apoptosis, promote angiogenesis and increase tumor cell proliferation [5]. Paracetamol (or acetaminophen) is a widely used non-NSAID analgesic that has a weak COX-inhibition effect [6]. Its action is essentially focused on COX-2 inhibition, while that of NSAIDs is mainly due to COX-1 inhibition. It is the major metabolite of phenacetin, an analgesic cataloged as carcinogenic by the International Agency for Research on Cancer (IARC) in 1987 and withdrawn from the market in most countries [7]. Both potential carcinogenic and chemopreventive effects of paracetamol have raised considerable interest in the last two decades [8,9]. In 1999, a report by the IARC found “inadequate evidence” of carcinogenicity for paracetamol both in human and animal studies, meaning that available data are insufficient to permit a conclusion regarding carcinogenicity [10].

Several epidemiological studies have examined the relation between paracetamol and hematologic malignancies. However, while some studies showed increased risk, others failed to do so [11]. Moreover, publication bias could have possibly distorted the global association between paracetamol and hematologic cancer [12]. Because of the considerable consumption of paracetamol worldwide, any association with increased or decreased risk of cancer may have important public health implications. Indeed, in the US, more than 600 over-the-counter and prescription medicines contain paracetamol [13], and in France, about half of the population uses this medicine [14].

The non-conclusive evidence of a relation between paracetamol and hematologic malignancies led us to conduct a meta-analysis of the studies published on the subject. This meta-analysis was registered in the PROSPERO database (ID: CRD42021245056).

## 2. Material and Methods

### 2.1. Eligibility Criteria and Search Strategy

We conducted a computerized literature search on Medline, EMBASE, WHO Global Index Medicus regional databases (AIM, LILACS, IMEMR, IMSEAR, WPRIM), from inception to December 2020. We used the following algorithm: (“acetaminophen”[MeSH Terms] OR “paracetamol”[All Fields]) AND (“hematologic neoplasms”[MeSH Terms] OR “leukemia”[MeSH Terms] OR “lymphoma”[MeSH Terms] OR “lymphoma”[All Fields] OR “Hodgkin”[All Fields] OR “multiple myeloma”[MeSH Terms] OR “myeloma”[All Fields]). We completed our search using a broader algorithm in free text words: (acetaminophen OR paracetamol) AND (neoplasms OR cancer).

We also searched meeting abstracts on the Conference Proceedings Citation Index- Science (CPCI-S) (January 1990 to December 2020) and Open Access Theses and Dissertations (OATD). We scanned the titles and abstracts of the identified articles to exclude those that were clearly irrelevant and examined the full text of the remaining articles. We contacted the authors if additional information on a report was needed. Furthermore, reference lists of original reports and narrative reviews were examined to identify additional relevant citations [12]. No limitation on language was set. Unpublished studies were not considered. The complete sequence of the search protocol is shown on Figure 1.

A study was included in the meta-analysis if it met the following criteria: (1) presented original data from case-control or cohort studies; (2) the exposure of interest was paracetamol intake; (3) the outcome of interest was a hematologic malignancy—leukemia, lymphoma or multiple myeloma; (4) presented relative risks (RR), odds ratios (OR), or standardized mortality or morbidity ratios (SMR) and their confidence intervals (CI), or enough data to calculate them (raw data, *p*-values, variance estimates). We assumed that the person-time of the unexposed group is much larger than that of the exposed group, and thus considered SMRs as equivalent to relative risks [15]. The difference between measures of association for mortality and incidence for the same cancer location and exposure is generally small [16]. We, therefore, pooled mortality and incidence studies together in a first analysis, and subsequently removed the only mortality study of our meta-analysis to check for robustness of our results.

One study that could have been relevant to our meta-analysis was excluded because it did not present any measure of RR or raw data that could be used. We contacted the authors for further information but were ultimately unable to obtain the data due to the fact that they were no longer available [17].

A study was defined as a specific cancer-exposure association. When effect measures of different malignancy types were reported in the same article (e.g., leukemia, lymphoma, etc.), we considered each cancer separately as a single study unit. However, in a subsequent analysis, for the same study we pooled the different malignancy estimates into one single measure.

We did not include studies that concerned childhood cancers related to the use of paracetamol by the mother. In studies with multiple publications from the same population, only data from the most recent publication were included.

### 2.2. Data Extraction

The following data were collected from each study in a standard extraction sheet: (1) author and year of publication; (2) study design; (3) number of cases and controls for case-control studies, and number of cases and sample size for cohort studies; (4) type of cancer divided into 4 main groups: all hematologic cancers, leukemia, lymphoma and multiple myeloma; (5) RR, OR or SMR and 95% CI for *any* exposure to paracetamol (any or unspecified dose or duration); (6) RR, OR or SMR and 95% CI for *high* exposure to paracetamol (highest dose or highest duration of intake provided by the authors); (7) variables used for adjustment, restriction or matching.

If different estimates were available, we used the one that was adjusted for the largest number of variables. Odds ratios from case-control studies were assumed to be unbiased estimates of the relative risk [18].

### 2.3. Quality Assessment

We assessed the quality of the studies using a 5-point quality score using elements of the Newcastle–Ottawa scale that were applicable to the setting of the present meta-analysis, and comprised characteristics of methods and presentation of results [19]. Specifically, for case-control studies, we determined whether participation rate was at least 80% in both groups; whether cases were incident (yes), prevalent, or dead (no); whether controls were taken from the general population (yes) or from one or various hospitals (no); whether potential confounding for sex, age, and smoking was corrected or prevented through matching or adjustment or whether the distribution of these factors was similar between index and comparison group; and whether duration of exposure to paracetamol was accurately measured. For cohort studies, in addition to the criteria listed above that were not specific to case-control designs, we determined whether loss to follow-up was less than 20% of the initial cohort size and whether efforts were made to ensure that the cohort did not change exposure during follow-up. When specific information on an item was not provided by the authors of the study, we scored this item as “no”. Studies with 4 or more points were classified as “low bias risk”, and the rest as “high bias risk”. The quality assessment was performed by 2 independent reviewers (J.P. and B.T.) and results were merged. Discrepancies were resolved by consensus.

The detailed score of each study is available in the Appendix A.

### 2.4. Statistical Analysis

We weighted the study-specific adjusted log ORs for case-control studies and log RRs or log SMRs for cohort studies by the inverse of their variance to compute a pooled OR and its 95% CI. We present both fixed and random effects pooled estimates but used the latter when heterogeneity was detected.

We used a bootstrap version of the DerSimonian and Laird Q test that was adapted to small samples to check for heterogeneity [20]. The null hypothesis of this test is absence of heterogeneity. To quantify this heterogeneity, we calculated the proportion of the total variance due to between-study variance (Ri statistic) [20].

We used funnel plots to assess publication bias visually. Because funnel plots have several limitations and represent only an informal way to detect publication bias, we carried out a more formal testing using the test proposed by Egger et al. [21]. To further evaluate and correct for publication bias, we used the trim-and-fill method [22].

### 2.5. Subgroup Analysis

In addition to the main analysis, we restricted the analysis to subgroups of studies defined by study characteristics such as case-control/cohort design, adjustment factors, and risk of bias category (low/high). These analyses were planned a priori and were performed with the software HEpiMA, version 2.1.3 [23], and Stata, version 12.0 (Stata Corp, College Station, TX, USA).

## 3. Results

We found 11 different articles that presented 17 distinct study units, carried out in 10 countries, on paracetamol use and hematologic malignancies that met the inclusion criteria and were finally included in the meta-analysis [24,25,26,27,28,29,30,31,32,33,34]. These included nine cohort studies and eight case-control studies. The studies and their main characteristics are shown in Table 1, Table 2, and Figure 2.

Our definition of “any intake” of paracetamol largely depended on the information provided by the individual studies. Four studies used paracetamol intake as a binary variable only (intake/no intake) [24,26,27,32]. Some studies provided information on cumulative dose under the form of tablet-years (n° of tablets/week for a n° of years) [29,30], while others used consumption frequency data [28,31], duration data [25] or a mixture of both [33,34]. “High intake”, when available, was defined as the highest quantity for which information was provided: highest cumulative dose [29,30], highest frequency [28,31], highest duration of use [25], or a mixture of frequency and duration [33,34]. One study used cumulative number of prescriptions >10 as an indicator of high intake [27].

In general, timing of exposure was not given in detail in the studies included in the meta-analysis. Some of the studies provided information on the lag time between first exposure to paracetamol and cancer ascertainment, a period of the cancer experience considered as unrelated to exposure [24,25,26,32,33].

We identified five studies on leukemia, three on multiple myeloma and seven on lymphoma (three on non-Hodgkin lymphoma, two on Hodgkin lymphoma and two on undetermined forms of lymphoma). We also found one study on chronic lymphocytic leukemia and small lymphocytic lymphoma that we finally included in the “all hematologic cancers” analysis [34], as it could not be classified into any leukemia or lymphoma categories [35]. The studies by Friis et al. and Walter et al. presented separate results for each malignancy [26,34]. The results for each malignancy were then considered separately in a first analysis. In a subsequent analysis, they were merged into one single estimate per article. The final results were very similar in both approaches.

Use of paracetamol was associated with increased odds of hematologic malignancies (pooled OR = 1.49, 95% CI = 1.23–1.80) (Table 3). The increase associated with high intake was more pronounced than that corresponding to any (or unspecified) use (pooled OR = 1.79, 95% CI = 1.55–2.06). Heterogeneity of effects was large for any intake (Ri = 0.79) but considerably smaller for high intakes (Ri = 0.41).

Both cohort studies and case-control studies show similar associations between any intake of paracetamol and hematologic malignancies. The association was slightly stronger for case-control studies (OR = 1.54, 95% CI = 1.19–1.99) than for cohort studies (OR = 1.43, 95% CI= 1.06–1.92) and heterogeneity was similarly high in both designs. For high intake of paracetamol, the association was stronger than that corresponding to any use. The pooled OR of the cohort studies was 1.83 (95% CI = 1.48–2.27) with no heterogeneity (Ri = 0.00) and that of case-control studies was 1.71 (95% CI = 1.21–2.41) with moderate heterogeneity (Ri = 0.65).

The group of studies with low risk of bias showed a stronger association than that of studies with high bias potential, both for any intake and high intake of paracetamol. Furthermore, the pooled odds ratio of fully adjusted studies (adjusted at least for age, sex and smoking) was larger than that of studies with incomplete adjustment: 1.56 (95% CI: 1.24–1.96) and 1.38 (95% CI 1.00–1.91), respectively.

Both any and high intakes of paracetamol were associated with increased odds for every hematologic malignancy except non-Hodgkin lymphoma. This increase was particularly high for multiple myeloma with a pooled OR of 2.13 (95% CI 1.51–3.0) for any intake and 3.16 (95% CI 1.96–5.10) for high intake. However, this analysis was based on two studies only.

The study by Lipworth et al. is the only study included in our meta-analysis that measured mortality and not incidence [27]. Also, an overlap of cases of this study with those of the prospective cohort study by Friis et al. that assessed incident cases is likely [26]. To assess the impact of this mortality study on our results, we performed a sensitivity analysis by excluding it from the main analysis. Except for a general decrease in heterogeneity, the results were not modified meaningfully, and our conclusions remained unaltered: pooled OR of “any intake” for all cancers = 1.43 (95% CI 1.20–1.71). No changes were observed for the high intake categories when this mortality study was excluded.

Regarding publication bias, while the funnel plot corresponding to high intake of paracetamol does not show any asymmetry, that corresponding to any intake is slightly skewed to the left (Figure 3), a feature confirmed by the Egger’s regression test (*p*-value = 0.001). However, in both intakes, the trim-and-fill analysis did not impute any additional study. To further evaluate the impact of publication bias in case-control studies, we recalculated the pooled estimates using the following extreme assumptions: (1) only half of the case-control studies ever conducted are published; (2) half of the unpublished studies found null associations (i.e., OR = 1); and (3) the average number of cases and control is similar in published and unpublished studies. Under these extreme assumptions, the pooled estimates would still be 1.27 (95% CI 1.09–1.49) for any intake and 1.34 (95% CI 1.13–1.59) for high intake.

## 4. Discussion

Our results show that paracetamol users are more likely to be diagnosed with hematologic malignancies. This association, observed in leukemia, multiple myeloma and lymphoma at large but not in non-Hodgkin lymphoma, is more pronounced for high intakes, both when all studies are taken together and when the analysis is performed by subgroup. As the studies included in this meta-analysis did not provide exact doses of paracetamol but, instead, simply a number of tablets consumed per day the dosage of which is unknown, we were not able to carry out a refined dose–response analysis. In particular, we could not compute effect measures for intakes >4 g/day, considered as a limit for toxicity. Cohort and case-control studies yielded similar results. Studies with low bias potential showed higher estimates than those with high bias potential. The effect observed is then unlikely to be due to a low quality of studies.

Several mechanisms could explain the association of paracetamol with hematologic malignancies: first, in vitro and in vivo studies have related paracetamol use to decreased DNA repair, and have shown that paracetamol may play the role of a co-mutagen [36]; second, the paracetamol metabolite N-acetyl-p-benzoquinone imine was shown to be a DNA topoisomerase II poison which was associated with secondary leukemia [37,38]. In addition, some experimental studies suggest that paracetamol is genotoxic to the bone marrow and could increase the risk of leukemia [39]; third, laboratory evidence showed that paracetamol increases NF-κB induction and affects IL-6 transcription. Both mechanisms possibly play a major role in the myeloma disease process. Chronic exposure to paracetamol may dysregulate these mechanisms [40]. However, one should bear in mind that subtypes of hematologic malignancies are etiologically heterogeneous. Furthermore, several methods are used to classify hematologic cancers by subtypes.

Our meta-analysis has some limitations. As data were not available in the original studies, we could not take into account possible interactions of paracetamol with other drugs, essentially opioid analgesics such as hydrocodone, a combination which is frequently dispensed in the US [41]. Furthermore, as explained above, due to the lack of data in the individual studies no detailed dose–response analysis was possible, besides that regarding high intake. We are aware that the assessment of consumption as any intake/high intake represents but a suboptimal way to measure the quantity of paracetamol ingested.

It is remarkable that, in several studies included in this meta-analysis, there was no mention of the time supposedly elapsed from intake of paracetamol to occurrence of the malignancy [24,27,28,29,30,31,32]. In other studies, the time excluded from follow-up, during which the occurrence of the event is impossible (“immortal” time) may have been too short [25,26].

Also, residual confounding may have distorted our results, as in any meta-analysis of observational studies. Although we are not aware of any genetic polymorphism that could play the role of confounder of the relation between paracetamol and hematologic cancer, it is possible that such a factor exists. As a matter of fact, recent studies discovered that a polymorphism of the COX-2 gene was a confounder of the NSAIDs–breast cancer relationship [42]. However, the existence of an unidentified factor, of genetic nature or else, associated with both exposure to paracetamol and hematologic malignancies, which could explain a high proportion of the observed effect, is unlikely. Even if this unidentified factor could double the risk of malignancy among subjects exposed to it (OR confounder-disease = 2) and, simultaneously, this factor happened to be twice more prevalent among subjects exposed to paracetamol than among non-exposed subjects (OR confounder-exposure = 2), the adjusted OR of the paracetamol–hematologic cancer relationship would still be 1.32 for any intake and 1.59 for high intake (assuming one-third of people are exposed to this unknown factor) [43]. Furthermore, our analysis of fully adjusted studies yielded a higher risk of malignancies than that of studies with incomplete adjustment.

Another limitation is the fact that paracetamol is particularly prone to “confounding by indication”, a kind of bias frequent in observational studies of drug exposure. This implies that the presence of some nonspecific symptoms (such as headaches or bone pain), that could be, in fact, early manifestations of the malignancy, could lead to intake of paracetamol, thus exaggerating its association with malignancies diagnosed subsequently. Furthermore, regular and heavy consumers of analgesics tend to have a higher level of comorbidity that can easily disturb the relationship between this and future health outcomes, if not properly accounted for. This kind of bias is more frequent in studies which use prevalent cases, or in studies of mortality without previous follow-up [44]. However, in the studies included in our meta-analysis, most case-control studies used incident cases, and the large majority adjusted for consumption previous to the diagnosis or to the first symptoms. Moreover, cohort studies, a design less prone to this kind of bias, as well as studies with low risk of bias showed a larger effect of paracetamol. The sensitivity analysis, performed after exclusion of the mortality cohort by Lipworth et al. [27], did not modify the results except for a decrease in heterogeneity.

Furthermore, there is some evidence of publication bias for the “any intake” group. This means that some relevant studies may not have been published and, therefore, were not included in the meta-analysis, which could potentially affect the results. This issue has been mentioned by previous narrative reviews [12]. In our meta-analysis, we did not include any limitation on language, year, or type of study. It is then unlikely that the search could miss relevant published studies. Also, the funnel plot shows a lack of studies at the right-hand side, i.e., studies that would present an increased risk of malignancies. This suggests that, in any case, our results are conservative, and the possible publication bias should go in the direction of an even more increased effect. Moreover, both the trim-and-fill method and the pooled estimates with extreme assumptions did not reverse our conclusions.

Previous studies show that 23% of the US population and 50% of the French population use paracetamol [14,45]. On the basis of these prevalences of use and our results, assuming that the associations we observed were of causal nature, we estimate that between 10% and 20% of hematologic malignancies may be attributable to paracetamol among users [46].

## 5. Conclusions

The magnitude of the associations, the consistency of the results through different settings, and the existence of a mechanism that gives biologic plausibility to the relationship, provide evidence that paracetamol may be associated with hematologic malignancies.

The fact that paracetamol is widely available over-the-counter in the majority of countries supposes a major public health issue that needs to be further investigated. More than measuring in a precise fashion the excess risk of hematologic malignancies among paracetamol users, our meta-analysis should be considered as a call for methodologically rigorous epidemiologic studies that would provide a definitive answer on the relation of paracetamol and hematologic cancers. In addition to assessing the dose–response relationship, these future studies should provide hypotheses on the lag time between exposure to paracetamol and occurrence of hematologic malignancies necessary for a causal relationship.

## Figures and Tables

**Figure 1 jcm-10-02429-f001:**
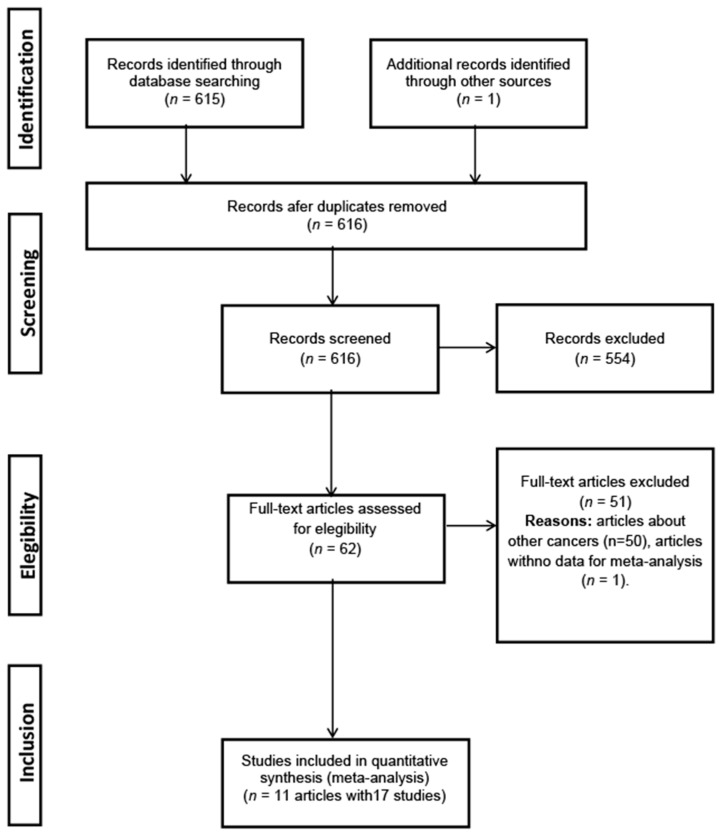
Flow chart of study selection.

**Figure 2 jcm-10-02429-f002:**
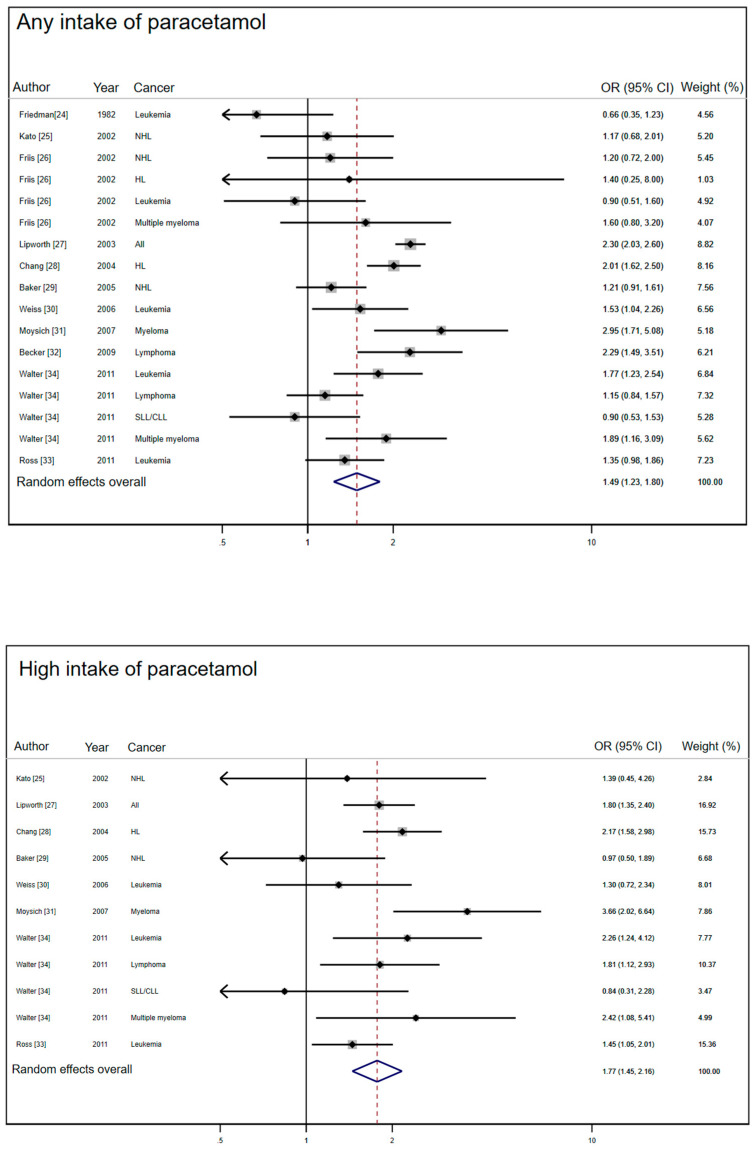
Forest plots of studies included in the meta-analysis of paracetamol intake and hematologic malignancies.

**Figure 3 jcm-10-02429-f003:**
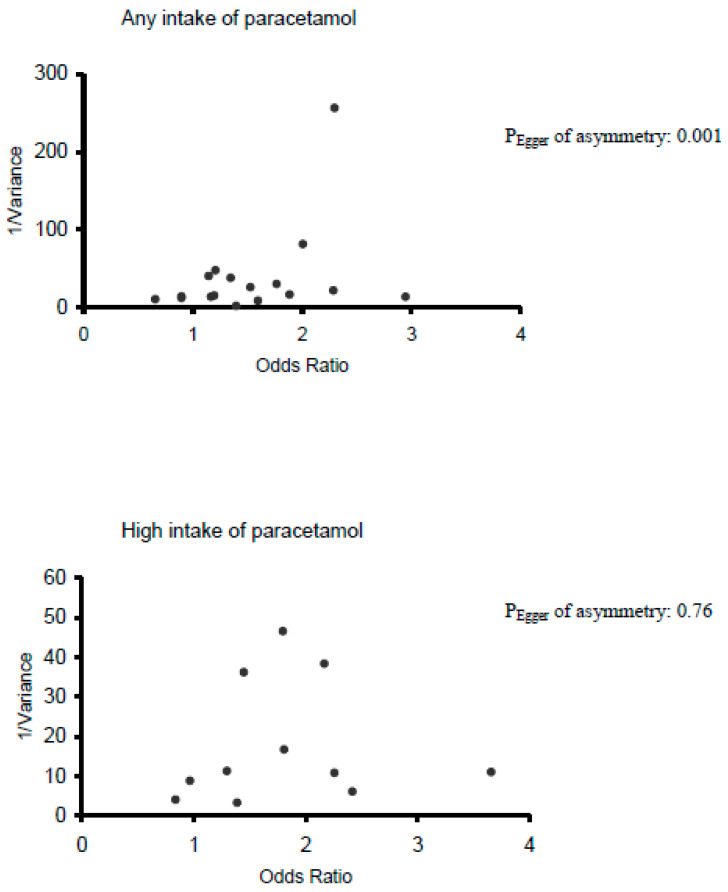
Funnel plots of Odds Ratio versus inverse variance of Odds Ratio.

**Table 1 jcm-10-02429-t001:** Characteristics of studies evaluating paracetamol intake and risk of hematologic malignancies.

Author	Study Period	Age of Participants (Years)	Paracetamol Intake Assessment	Cancer Ascertainment	Lag Time ^a^	Definition of Any Intake of Paracetamol	Definition of High Intake of Paracetamol
Friis et al. [26]	1989–1995	63 (average)	Regional prescription database	Cancer registry	1 year	Prescribed/Not prescribed	-
Lipworth et al. [27]	1989–1995	>16	Regional Pharmacoepidemiologic database	National mortality files	None	SMR based on prescribed/not prescribed	>10 prescriptions
Walter et al. [34]	2000–2002	50–76	Self-administered questionnaire	Cancer registry	Not given	Pooling of OR corresponding to<4 days/week or <4 years, and ≥4 days/week or ≥4 years	≥4 days/week or ≥4 years
Friedman et al. [24]	1971–1976	>30	Hospital charts	Hospital charts	6 months	Intake/no intake	-
Kato et al. [25]	1995–1998	20–79	Telephone interview	Cancer registry	1 year	Pooling of OR corresponding to <3 years of use, 3–10 years and >10 years	Consumption for >10 years
Chang et al. [28]	1997–2000	15–79	Telephone interview	Hospital charts and Cancer registry	None	Pooling of OR corresponding to <2 tablets/week and ≥2 tablets/week	≥2 tablets/week
Baker et al. [29]	1982–1998	57 (average)	Self-administered questionnaire	Cancer registry	None	Pooling of OR corresponding to ≤10 tablet-year ^b^ and >10 tablet-year	>10 tablet-year
Weiss et al. [30]	1981–1998	20–84	Self-administered questionnaire	Hospital cases	Not given	Pooling of OR corresponding to ≤4 tablet-year and >4 tablet-year	>4 tablet-year
Moysich et al. [31]	1982–1998	60 (average)	Self-administered questionnaire	Hospital cases	Not given	Pooling of OR corresponding to <7 times/week and ≥7 times/week	≥7 times/week
Becker et al. [32]	1998–2004	<71	Self-administered questionnaire	Different hospitals	3 years	Intake/no intake	-
Ross et al. [33]	2005–2009	20–79	Self-administered questionnaire	Regional cancer surveillance system	2 years	OR corresponding to ≥1use/week for ≥1 year	Pooling of OR corresponding to ≥7 tablets/week and >10 years

^a^: Lag time: minimum time elapsed between first intake of paracetamol and participation in the study, ^b^: 1 tablet-year = 1tablet/day * years of use. Abbreviations: SMR, Standardized Mortality Ratio; OR, odds ratio.

**Table 2 jcm-10-02429-t002:** Odds ratios of paracetamol intake and hematologic malignancies in individual studies.

Author	Year	Country	Sample Size/#Cases	Cancer	RR (95% CI)Any Intake (a)	RR (95% CI)High Intake (b)	Adjustment/Matching
Cohort studies							
Friis et al. [26]	2002	Denmark	39946/46	Leukemia	0.90 (0.50–1.60)	-	Age, sex, other analgesic use
Friis et al. [26]	2002	Denmark	39946/25	Multiple myeloma	1.60 (0.60–3.20)	-	Age, sex, other analgesic use
Friis et al. [26]	2002	Denmark	39946/47	NHL	1.20 (0.70–2.00)	-	Age, sex, other analgesic use
Friis et al. [26]	2002	Denmark	39946/6	HL	1.40 (0.00–8.00)	-	Age, sex, other analgesic use
Lipworth et al. [27]	2003	Denmark	49890/286	All cancers	2.30 (2.00–2.60)	1.80 (1.40–2.40)	Age, sex
Walter et al. [34]	2011	USA	64839/66	Leukemia	1.77 (1.24–2.54)	2.26 (1.24–4.12)	Age, sex, education, ethnicity, smoking, medical history
Walter et al. [34]	2011	USA	64839/235	Lymphoma	1.15 (0.85–1.57)	1.81 (1.12–2.93)	Age, sex, education, ethnicity, smoking, medical history
Walter et al. [34]	2011	USA	64839/88	SLL/CLL	0.90 (0.53–1.53)	0.84 (0.31–2.28)	Age, sex, education, ethnicity, smoking, medical history
Walter et al. [34]	2011	USA	64839/136	Multiple myeloma	1.89 (1.16–3.09)	2.42 (1.08–5.41)	Age, sex, education, ethnicity, smoking, medical history
Case-control studies							
Friedman et al. [24]	1982	USA	409/818	Leukemia	0.66 (0.35–1.23)	-	Age, sex
Kato et al. [25]	2002	USA	376/463	NHL	1.17 (0.68–2.01)	1.39 (0.45–4.26)	Age, sex, education, body mass index, history of hematologic cancer, study year
Chang et al. [28]	2004	USA	565/679	HL	2.01 (1.62–2.50)	2.17 (1.58–2.98)	Age, sex, residence, smoking, other analgesic use
Baker et al. [29]	2005	USA	628/2512	NHL	1.21 (0.90–1.61)	0.97 (0.49–1.89)	Age, sex, race, study year, smoking, education, income.
Weiss et al. [30]	2006	USA	169/676	Leukemia	1.53 (1.03–2.26)	1.30 (0.73–2.34)	Age, sex, race, study year, smoking, education, alcohol consumption
Moysich et al. [31]	2007	USA	117/483	Myeloma	2.95 (1.72–5.08)	3.66 (2.02–6.64)	Age, smoking, study year
Becker et al. [32]	2009	Europe	2362/2465	Lymphoma	2.29 (1.49–3.51)	-	Age, sex, study center.
Ross et al. [33]	2011	USA	670/701	Leukemia	1.35 (0.98–1.86)	1.45 (1.04–2.01)	Age, sex, body mass index, other analgesic use

(a) Any intake: any (or unspecified) dose or duration of paracetamol intake, (b) High intake: highest dose or longest duration of paracetamol intake available in the individual study. Abbreviations: RR, relative risk; CI, confidence interval; NHL, non-Hodgkin lymphoma; HL, Hodgkin lymphoma; CLL, chronic lymphocytic leukemia; SLL, small lymphocytic lymphoma.

**Table 3 jcm-10-02429-t003:** Pooled relative risks (RR) and 95% confidence intervals (CI) of paracetamol intake and hematologic malignancies.

Any Intake ^(b)^	Number of Studies	RR (95% CI) Fixed Effects	RR (95% CI) Random Effects	Ri ^(a)^	Q Test *p*-Value
All studies	17	1.78 (1.64–1.92)	1.49 (1.23–1.80)	0.79	0.001
Cohort studies	9	1.90 (1.72–2.09)	1.43 (1.06–1.92)	0.85	0.001
Case-control studies	8	1.93 (1.74–2.14)	1.54 (1.19–1.99)	0.74	0.001
Low bias risk	6	1.99 (1.82–2.19)	1.66 (1.27–2.16)	0.85	0.001
High bias risk	11	1.39 (1.21–1.59)	1.39 (1.11–1.73)	0.56	0.01
Fully adjusted	8	1.58 (1.40–1.78)	1.56 (1.24–1.96)	0.70	0.001
Incompletely adjusted	9	1.60 (1.41–1.82)	1.38 (1.00–1.91)	0.87	0.001
By histologic type:					
Leukemia	5	1.34 (1.12–1.61)	1.26 (0.93–1.69)	0.59	0.04
Lymphoma (all)	7	1.54 (1.35–1.76)	1.46 (1.14–1.89)	0.67	0.001
Non-Hodgkin	3	1.20 (0.96–1.51)	1.20 (0.96–1.51)	0.00	0.99
Hodgkin	2	2.00 (1.61–2.48)	2.00 (1.61–2.48)	0.00	0.68
Undetermined	2	1.46 (1.14–1.88)	1.60 (0.81–3.13)	0.86	0.01
Multiple Myeloma	3	2.13 (1.54–2.94)	2.13 (1.51–3.01)	0.12	0.32
High Intake ^(c)^	Number of Studies	RR (95% CI) Fixed Effects	RR (95% CI) Random Effects	Ri ^(a)^	Q Test *p*-Value
All studies	11	1.79 (1.55–2.06)	1.77 (1.45–2.16)	0.41	0.08
Cohort studies	5	1.83 (1.48–2.27)	1.83 (1.48–2.27)	0.00	0.51
Case-control studies	6	1.76 (1.45–2.12)	1.71 (1.21–2.41)	0.65	0.02
Low bias risk	6	1.93 (1.62–2.31)	1.93 (1.62–2.31)	0.00	0.54
High bias risk	5	1.56 (1.24–1.98)	1.59 (1.03–2.44)	0.66	0.03
Fully adjusted	8	1.93 (1.59–2.33)	1.84 (1.38–2.47)	0.52	0.04
Incompletely adjusted	3	1.63 (1.32–2.01)	1.63 (1.32–2.01)	0.00	0.59
By histologic type:					
Leukemia	3	1.54 (1.19–1.99)	1.54 (1.19–2.00)	0.01	0.36
Lymphoma (all)	4	1.83 (1.44–2.33)	1.70 (1.20–2.40)	0.44	0.18
Non-Hodgkin	2	1.07 (0.60–1.89)	1.07 (0.60–1.89)	0.00	0.58
Multiple Myeloma	2	3.16 (1.96–5.10)	3.16 (1.96–5.10)	0.00	0.41

^(a)^ Ri: Proportion of total variance due to between-study variance, ^(b)^ Any intake: Any dose or duration of paracetamol intake, ^(c)^ High intake: highest dose or longest duration of paracetamol intake available in the individual studies.

## Data Availability

Any data necessary to replicate the analysis of this article is available from the corresponding author.

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
