# Peer review of "Paracetamol Intake and Hematologic Malignancies: A Meta-Analysis of Observational Studies"

_jcm, 2021, doi:10.3390/jcm10112429_

Round 1

Reviewer 1 Report

This is an interesting and well written paper but it has one large deficit which I feel needs to be corrected.

There needs to be a discussion of the definition of of "any" and "high" intake. As well as a discussion of how the studies ascertained the data.

This is the exposure of interest and is entirely glossed over. I understand that there is a large range in the studies but this needs to be explored and explained. The implication of "any" use is that a person took the drug of interest at any point but was it any point in the week, month, year before the diagnosis. As the authors point out if it was near the diagnosis it may be related to the disease but if it was a year before that would not hold.

I would suggest that the table that already exists should be expanded to include the "any" and "high" dosages for each study and that this should be discussed/explained in the text.

Author Response

Comment: This is an interesting and well written paper but it has one large deficit which I feel needs to be corrected.  There needs to be a discussion of the definition of of "any" and "high" intake. As well as a discussion of how the studies ascertained the data. I would suggest that the table that already exists should be expanded to include the "any" and "high" dosages for each study and that this should be discussed/explained in the text.

Answer: We thank the reviewer for this comment and the suggestion to add information. However, instead of adding information to the existing table that was already very dense, we preferred to add a new table with the following information: study period, age of participants, Paracetamol intake assessment, cancer assessment, lag time between paracetamol consumption and participation in the study, definition of any intake of paracetamol, and definition of high intake of paracetamol. This table is now Table 1 and its title is: “Characteristics of studies evaluating paracetamol intake and risk of hematologic malignancies”, while former Table 1 is now Table 2 and harbors the title of: “Odds ratios of paracetamol intake and hematologic malignancies in individual studies”.

We also added the following text in the results section: “Our definition of “any intake” of paracetamol largely depended on the information provided by the individual studies. Four studies used paracetamol intake as a binary variable only (intake/no intake)[22, 24, 25, 30]. Some studies provided information on cumulative dose under the form of tablet-years (nº of tablets/week for a nº of years) [27-28], while others used consumption frequency data [26, 29], duration data [23] or a mixture of both [31-32].” “High intake”, when available, was defined as the highest quantity for which information was provided: highest cumulative dose [27, 28], highest frequency [26, 29], highest duration of use [23], or a mixture of frequency and duration [31, 32]. One study used cumulative number of prescription >10 as an indicator of high intake [25].   

Comment: This is the exposure of interest and is entirely glossed over. I understand that there is a large range in the studies but this needs to be explored and explained. The implication of "any" use is that a person took the drug of interest at any point but was it any point in the week, month, year before the diagnosis. As the authors point out if it was near the diagnosis it may be related to the disease but if it was a year before that would not hold.

Answer: The individual studies provided much less information than expected. There is no timing of exposure to paracetamol. Indeed, some studies just provide this information as a binary variable “consumed parecetamol/ did not consume paracetamol. However, some of the studies included in the meta-analysis provide information on the lag time between first exposure to paracetamol and cancer ascertainment. We provide these details in our new table 1. Furthermore, in the results section we now say “In general, timing of exposure was not given in detail in the studies included in the meta-analysis. Some of the studies provided information on the lag time between first exposure to paracetamol and cancer ascertainment, a period of the cancer experience considered as unrelated to exposure [22-24, 30, 31].”    

In the discussion, we had already written the following sentences in the first version of the manuscript: “It is remarkable that, in several studies included in this meta-analysis, there was no mention of the time supposedly elapsed from intake of paracetamol to occurrence of the malignancy [22, 25-30]. In other studies, the time excluded from follow-up, during which the occurrence of the event is impossible (“immortal“time) may have been too short [23, 24].”  

Reviewer 2 Report

The authors undertook a task of conducting this metanalysis to access the association of acetaminophen with hematologic malignancies. The studies that were finally incorporated into the analysis were very heterogeneous likely because of the use of similar databases especially in the case of the Denmark and USA based cohort studies. The analysis further fails to describe the dosage and duration of acetaminophen exposure because of the lack of reported data and can’t account for off the counter use of the medication in the control arms. These limitations are addressed to a degree in the discussion section of the study. The study seems of sound statistical and methodological design.  

Minor recommendations

In the introduction consider differentiating NSAIDS COX inhibition focused on COX-1 vs acetaminophen which most of its inhibition is COX-2.

Page 1 Line 2: Consider using the generic acetaminophen instead of paracetamol in the title and text.

Page 1 Line 38 the sentence “. Paracetamol (or acetaminophen) is a widely used non-NSAID 38 analgesic that has a weak COX-inhibition effect” lacks a reference.

Author Response

Comment: The authors undertook a task of conducting this metanalysis to access the association of acetaminophen with hematologic malignancies. … The study seems of sound statistical and methodological design

Answer: Thank you for this comment. We appreciate it.

Comment: Minor recommendations. In the introduction consider differentiating NSAIDS COX inhibition focused on COX-1 vs acetaminophen which most of its inhibition is COX-2.

Answer: We have added the following sentence in the introduction: “Its action is essentially focused on COX-2 inhibition, while that of NSAIDs is mainly due to COX-1 inhibition.” 

Comment: Page 1 Line 2: Consider using the generic acetaminophen instead of paracetamol in the title and text.

Answer: We would like to mention that Reviewer 3 requests exactly the opposite: to refrain from using acetaminophen and instead use paracetamol. We do not have any preference and will adapt to what the Editor wishes if the manuscript is finally accepted for publication. The term paracetamol is as frequently used as acetaminophen. There is a tendency to use acetaminophen more frequently in the US, while paracetamol is preferred in Europe.  

Comment: Page 1 Line 38 the sentence “. Paracetamol (or acetaminophen) is a widely used non-NSAID analgesic that has a weak COX-inhibition effect” lacks a reference.

Answer: We have added the following reference: Catella-Lawson F et al. Cyclooxygenase inhibitors and the antiplatelet effects of aspirin. N Engl J Med. 345:1809-1817 (2001) that says ”We found that, at a dose of 1000 mg, acetaminophen is a weak, reversible, isoform-nonspecific cyclooxygenase inhibitor”.

Reviewer 3 Report

This paper is a meta-analysis of epidemiologic studies of paracetamol and hematologic cancers. It is well written, and the methods are highly appropriate and comprehensive. I have no major concerns.

My main comments are:

  1. Hematologic cancers are etiologically heterogeneous. This should be mentioned in the introduction and discussion. The Discussion should mention both that there are different etiologies by subtypes of hematologic malignancies and also varying ways used to classify the subtypes.
  2. In a related concern, the separate results by type of hematologic cancer should be given more emphasis. Specifically, there should be a second sentence after the one sentence of results in the Abstract, such as “ “By subtype, risk was higher for MM (2.13 (give CI) for any use and 3.16 (give CI) for high intake, while risk was lower and non-significant for NHL.”
  3. The Methods page 3 say the exposure of interest is paracetamol intake. But Methods page 4 states ” “when both data on global analgesic use including paracetamol and data on exclusive use of paracetamol were available for the same study, we used the later.” This sentence implies that when data on only total use of analgesics were reported, the authors used total analgesic use as the exposure. That would be invalid, and instead those studies must be excluded. The authors need to confirm that they did not use total analgesic use as an exposure for any study in their response. In that case, they only need to change (or omit) that sentence to make their methods clearer. If they did use total analgesic use as a proxy for paracetamol use, this needs to be justified by the percentage of total analgesic use that is paracetamol in each country studied and note for which studies the proxy was used.

Minor concerns:

  1. Use of “acetaminophen “ vs “paracetamol.” Acetaminophen should be used in the abstract in parentheses after paracetamol or added to keywords, so that researchers can find this article. But in the paper, the authors should stick with paracetamol. Specifically figure 2 uses acetaminophen.
  2. Page 8. Paragraph that starts with “One study” seems to be about a study not included in the results. Then that paragraph should be in methods, not results.
  3. Page 8 states one study used the category CLL/SLL “which did not fit into any WHO category.” This is not correct. CLL/SLL is a subcategory of the WHO category “Mature B-cell neoplasms” in both 2008 and 2016 versions. It seems the authors are using ICD categories not WHO categories of hematologic malignancies. So, the authors should instead say one study used the WHO CLL/SLL category which they could not classify into either their leukemia or lymphoma categories.
  4. Figure 2 legend should give the exposure and outcome used in the figure.
  5. Table 1 format-there should be only one table title, but 2 subheadings –cohort and case-control.
  6. Figure 3 is not needed. The contents of the figure are already explained in the Results text (or they could be expanded if the authors feel they need to be).
  7. Discussion says that paracetamol plus hydrocodone is the combination most frequently dispensed in the US. That seems misleading. Maybe that is the most common medical prescription use of paracetamol, but I would think the large majority of paracetamol use in the US is paracetamol use alone, purchased over the counter (non-prescription).

Author Response

Comment: This paper is a meta-analysis of epidemiologic studies of paracetamol and hematologic cancers. It is well written, and the methods are highly appropriate and comprehensive.

Answer: Thank you for this comment. We appreciate it

Comment: Hematologic cancers are etiologically heterogeneous. This should be mentioned in the introduction and discussion. The Discussion should mention both that there are different etiologies by subtypes of hematologic malignancies and also varying ways used to classify the subtypes.

Answer: In the Introduction we say: “Hematologic malignancies are a heterogeneous group of diseases of diverse incidence, prognosis, and etiology.” In the Discussion we have added the following sentence: “However, one should bear in mind that subtypes of hematologic malignancies are etiologically heterogeneous. Furthermore, several methods are used to classify hematologic cancers by subtypes.” 

Comment: In a related concern, the separate results by type of hematologic cancer should be given more emphasis. Specifically, there should be a second sentence after the one sentence of results in the Abstract, such as “ “By subtype, risk was higher for MM (2.13 (give CI) for any use and 3.16 (give CI) for high intake, while risk was lower and non-significant for NHL.”

Answer: We have added what the reviewer suggested and now say: “By subtype, risk was higher for multiple myeloma (OR 2.13, 95% CI 1.54-2.94) for any use and OR 3.16, 95% CI 1.96-5.10 for high intake, while risk was lower and non-significant for non-Hodgkin lymphoma.

Comment: The Methods page 3 say the exposure of interest is paracetamol intake. But Methods page 4 states ” “when both data on global analgesic use including paracetamol and data on exclusive use of paracetamol were available for the same study, we used the later.” This sentence implies that when data on only total use of analgesics were reported, the authors used total analgesic use as the exposure. That would be invalid, and instead those studies must be excluded. The authors need to confirm that they did not use total analgesic use as an exposure for any study in their response. In that case, they only need to change (or omit) that sentence to make their methods clearer. If they did use total analgesic use as a proxy for paracetamol use, this needs to be justified by the percentage of total analgesic use that is paracetamol in each country studied and note for which studies the proxy was used.

 Answer: We are sorry if this sentence was misleading. Of course, we never used total analgesic in our analysis. We used paracetamol exclusively. If a study had data on total analgesic only, it was not even further considered. As suggested by the reviewer, we deleted the sentence.

Comment: Minor concerns: Use of “acetaminophen “ vs “paracetamol.” Acetaminophen should be used in the abstract in parentheses after paracetamol or added to keywords, so that researchers can find this article. But in the paper, the authors should stick with paracetamol. Specifically figure 2 uses acetaminophen.

Answer: We have followed this advice but we would like to highlight that Reviewer 2 suggested exactly the opposite: refrain from using paracetamol and use acetaminophen instead. We do not mind using any of the terms, and therefore, we leave the choice open to the Editor, should the manuscript be accepted for publication. In general both terms are equally used in the scientific literature. Acetaminophen is used more frequently in the US, whole paracetamol is preferred in Europe.

Comment: Page 8. Paragraph that starts with “One study” seems to be about a study not included in the results. Then that paragraph should be in methods, not results.

Answer: We have moved the paragraph to methods.

Comment: Page 8 states one study used the category CLL/SLL “which did not fit into any WHO category.” This is not correct. CLL/SLL is a subcategory of the WHO category “Mature B-cell neoplasms” in both 2008 and 2016 versions. It seems the authors are using ICD categories not WHO categories of hematologic malignancies. So, the authors should instead say one study used the WHO CLL/SLL category which they could not classify into either their leukemia or lymphoma categories.

Answer: We are sorry for our lack of clarity concerning this issue. Indeed, what we meant was that CLL/SLL could not be classified into any leukemia or lymphoma categories. We have rephrased the sentence consequently.

Comment: Figure 2 legend should give the exposure and outcome used in the figure.

Answer: The legend has been modified to “Forest plots of studies included in the meta-analysis of paracetamol intake and hematologic malignancies.”

Comment: Table 1 format-there should be only one table title, but 2 subheadings –cohort and case-control.

Answer: We have restructured the tables as suggested. Please note that Table 1 is now Table 2.

Comment: Figure 3 is not needed. The contents of the figure are already explained in the Results text (or they could be expanded if the authors feel they need to be).

Answer: We have no problem in deleting this figure. However, we believe that the description made in the results “… while the funnel plot corresponding to high intake of paracetamol does not show any asymmetry, that corresponding to any intake is slightly skewed to the left” and the comment in the Discussion “There is some evidence of publication bias for the “any intake” group. Also, the funnel plot shows a lack of studies at the right hand side” imply that the reader, in case the manuscript is accepted for publication, is observing the figure that is described and commented. It is difficult to understand what “skewed to the left” and “the right hand side” mean if there is no figure. We would appreciate it if the reviewer could confirm their choice of deleting this figure, in which case we would simply write “data not shown” instead of “fig. 3”.

Comment: Discussion says that paracetamol plus hydrocodone is the combination most frequently dispensed in the US. That seems misleading. Maybe that is the most common medical prescription use of paracetamol, but I would think the large majority of paracetamol use in the US is paracetamol use alone, purchased over the counter (non-prescription).

Answer: We agree that the paracetamol presentation most frequently used is paracetamol alone, as an over-the-counter medicine. However, our point was to mention the frequent prescription of combinations of medicines which contain paracetamol and opioids. Our comment was based on the appendix “Top Medicines by Prescription” p49 of the IQVIA National Prescription Audit, Jan 2018 [https://www.iqvia.com/-/media/iqvia/pdfs/institute-reports/medicine-use-and-spending-in-the-us-a-review-of-2017-and-outlook-to-2022.pdf?_=1619639285637], in which we observe that, in 2013, this combination was the most prescribed in the US. To avoid any misunderstanding we modified the sentence and now say:  “…we could not take into account possible interactions of paracetamol with other drugs, essentially opioid analgesics such as hydrocodone, a combination which is frequently dispensed in the US [40].”

Round 2

Reviewer 1 Report

Given the very short to no lag time in the use of paracetamol it is possible that all of the subjects already had a malignancy when they started taking the drug - entirely invalidating the results. It is exceedingly unlikely in my opinion that taking anything for a year or six months, or shorter, as is implied by the authors of the original papers could lead to the development of malignancies and it is scientifically inappropriate to suggest so. It introduce a bias which is not addressed in the paper. In addition there are 3 studies included that use prescriptions when this is a readily available over the counter drug which is very commonly and frequently used.   I think a less aggressive conclusion and more discussion of the flaws of the original papers might make it more appropriate. 

Author Response

Comment: Given the very short to no lag time in the use of paracetamol it is possible that all of the subjects already had a malignancy when they started taking the drug - entirely invalidating the results. It is exceedingly unlikely in my opinion that taking anything for a year or six months, or shorter, as is implied by the authors of the original papers could lead to the development of malignancies and it is scientifically inappropriate to suggest so. It introduce a bias which is not addressed in the paper. In addition there are 3 studies included that use prescriptions when this is a readily available over the counter drug which is very commonly and frequently used.   I think a less aggressive conclusion and more discussion of the flaws of the original papers might make it more appropriate.

Answer: We believe that our risk of bias assessment was reasonable and followed widely supported scales. No epidemiologic study is absolutely free of bias, not even randomized (experimental) studies. Ideally, a lag time between exposure and outcome should be proposed. However, its duration is arbitrary no matter how good the design of the study is. The first studies about tobacco smoking and lung cancer, which represent the birth of modern epidemiology, did not have any hypothesis on the lag time necessary for tobacco to cause lung cancer. Furthermore, even the most knowledgeable cohort studies such as the Nurses’ Health Study, don’t set a minimum lag time between exposure and outcome occurrence, i.e. these studies do not directly eliminate those cases of disease that occur in a certain time after exposure. As a second step, they may perform an analysis stratified in different lag time periods.

The fact that the studies included in our meta-analysis do not wait years or decades after paracetamol prescription to start collecting cases of hematologic malignancies does not imply that these studies are flawed. We are aware that some cases of the disease, if there is no lag time, may occur during a period of exposure considered as non-relevant. Indeed, from the first version of the manuscript we had mentioned this issue in the discussion by saying: “It is remarkable that, in several studies included in this meta-analysis, there was no mention of the time supposedly elapsed from intake of paracetamol to occurrence of the malignancy [24, 27-32]. In other studies, the time excluded from follow-up, during which the occurrence of the event is impossible (“immortal“ time) may have been too short [25, 26].” We also emphasized this issue in the last sentence of the paper, as a conclusion: “Future studies should assess the dose-response relationship and define a lag time between exposure and outcome measurement that allows for a plausible latency”.  

However, there is no reason to believe that the majority of malignancy cases of the studies included in our meta-analysis occur within days or weeks following first intake of paracetamol. These “useless” cases are scarce. Also, paracetamol is largely consumed over-the-counter. It is highly probable that all the subjects included in the individual studies of our meta-analysis did not start consuming paracetamol when prescribed, but probably much earlier, for banal motives such as common cold, fever o toothache.        

Furthermore, this and other risks of bias have been assessed in the quality scoring and the corresponding analyses have been carried out. We do not believe that an insufficient definition of the lag time is enough to dismiss the results of a study altogether and to label this study as inappropriate. The studies of this meta-analysis were published in extremely selective journals and demanding in methodologic quality. (Int J Cancer, J Clin Epidemiol, J Nat Cancer Inst, Cancer Cause Control, J Chronic Dis etc.).

Our Discussion has an extension of 1038 words. 680 of them, i.e. two-thirds, concern the discussion of the limitations of the meta-analysis. We believe it is difficult to be more self-critical than what we have been in this article.

Paracetamol represent a ubiquitous exposure and the outcome (hematologic malignancies) is a severe one. Unfortunately, methodologically perfect studies on this relation do not exist and probably never will, as randomized studies are probably not feasible since they would demand an extremely long follow-up. We believe we cannot hide the results of this meta-analysis, just because the studies included in it are observational and some of them did not give precise information on the lag time.

To add even more caution to our study, we have appended the following paragraph to the new version of the article, as a general conclusion: “More than measuring in a precise fashion the excess risk of hematologic malignancies among paracetamol users, our meta-analysis should be considered as a call for methodologically rigorous epidemiologic studies that would provide a definitive answer on the relation of paracetamol and hematologic cancers. In addition to assessing the dose-response relationship, these future studies should provide hypotheses on the lag time between exposure to paracetamol and occurrence of hematologic malignancies necessary for establishing a causal relationship.

We would be glad to add any sentence in that sense, should the reviewer or the Editor have any suggestion.